

# New methods for the calibration of optical resonators: Integrated Calibration by means of Optical Modulation (ICOM) and Narrow Band Cavity Ring-Down (NB-CRD)

Henning Finkenzeller[1], Denis Pöhler[1,2], Martin Horbanski[1,2], Johannes Lampel[1,2], and Ulrich Platt[1,2]

[1]Institute of Environmental Physics, University of Heidelberg, INF 229, 69120 Heidelberg, Germany
[2]Airyx GmbH, Eppelheim, Germany

**Correspondence:** Henning Finkenzeller (henning.finkenzeller@posteo.de)

**Abstract.** Optical resonators are used in spectroscopic measurements of atmospheric trace gases to establish long optical path lengths $L(\lambda)$ with enhanced absorption in compact instruments. In cavity-enhanced broad band methods, the exact knowledge of both the magnitude of $L(\lambda)$ and its spectral dependency is fundamental for the correct retrieval of trace gas concentrations. $L(\lambda)$ is connected to the spectral mirror reflectivity $R(\lambda)$ which is often referred to instead. $L(\lambda)$ is also influenced by other quantities like broad-band absorbers or alignment of the optical resonator. The established calibration techniques to determine $L(\lambda)$, e.g., introducing gases with known optical properties or measuring the ring-down time, all have limitations: Limited spectral resolution, insufficient absolute accuracy and precision, inconvenience for field deployment, or high cost of implementation. Here, we present two new methods that aim to overcome these limitations: (1) Narrow Band Cavity Ring-Down (NB-CRD) uses Cavity Ring-down and a tunable filter to retrieve spectrally resolved path lengths. (2) Integrated Calibration by means of Optical Modulation (ICOM) allows the determination of the optical path-length at the spectrometer resolution with high accuracy in a relatively simple setup. In a prototype set-up we demonstrate the high accuracy and precision of the new approaches. The methods facilitate and improve the determination of $L(\lambda)$, thereby simplifying the use of cavity enhanced absorption spectroscopy.

## 1 Introduction

Spectroscopic measurements of atmospheric trace gases are based on the extinction (i.e., scattering and absorption) of light along its propagation: $I_0(\lambda) \rightarrow I(\lambda)$. The degree of attenuation depends on the particular gas concentrations $c_i$, the absorption cross sections $\sigma_i$, the scattering coefficient of air $\sigma_{\mathrm{scat}}$ and the light path length $L(\lambda)$. Especially for only weakly absorbing gases, long light paths are required to enhance the extinction. Multi-pass spectroscopic absorption cells employed mirrors to enhance the path length in laboratory setups by a factor of up to several 10 (e.g. Pfund, 1939; White, 1942; Herriott and Schulte, 1965; Thoma et al., 1994). More recently, cavity-enhanced resonating designs using concave dichroic mirrors have become feasible (Zheng et al., 2018), in which the light undergoes a non-geometrical number of reflections (O'Keefe and Deacon, 1988; Berden and Engeln, 2009). The path length enhancements established with this technique can routinely exceed a factor of $> 10^4$. It is a characteristic property of broad-band cavity enhanced absorption spectroscopy (BB-CEAS) cavity enhanced



or differential optical absorption spectroscopy (CE-DOAS) that the length of the light path $L(\lambda)$ is usually strongly wavelength

dependent (e.g. He et al., 2020) — for most other differential optical absorption spectroscopy (DOAS, (Platt and Stutz, 2008)) variants the light path is independent or only weakly varying with wavelength. Once $L(\lambda)$ and the material properties $\sigma_i$ are known, conclusions about the concentrations $c_i$ can be drawn from measurements of $I(\lambda)$ and $I_0(\lambda)$ as e.g. shown in Zhu et al. (2018); Tang et al. (2020). The spectral shape of $L(\lambda)$ modulates the strength of absorption features and needs to be known to unambiguously attribute observed absorption features to particular absorbers and determine their concentrations (Platt et al.,

2009; Horbanski et al., 2019). To not limit the accuracy of the retrieved trace gas concentration, the path length accuracy should be better than the uncertainty in the absorption cross sections, which commonly reaches few percent.

In optical resonators, $L(\lambda)$ cannot be determined through sole geometrical considerations. It is given by the quality of the resonator, predominately the mirror separation, the mirror reflectivity, the Rayleigh extinction of the sample gas, and the aerosol load in the measurement cell (Fiedler et al., 2007; Platt et al., 2009; Thalman and Volkamer, 2010; Wang et al., 2015):

$$L(\lambda) = \frac{d_0}{1 - R(\lambda) - \epsilon(\lambda)} \qquad (1)$$

where $d_0$ is the distance between the two mirrors, $R(\lambda)$ the mirror reflectivity, and $\epsilon(\lambda)$ the extinction for a pass in the measurement cell. The determination of $L(\lambda)$ is therefore closely connected to the determination of the mirror reflectance $R(\lambda)$, which is also often used as characterising parameter. At typical mirror reflectances close to unity (typically $R > 0.999$), the mirrors can not be cleaned and prepared to a sufficiently reproducible state. Consequently, $L(\lambda)$, or $R(\lambda)$, varies between

measurement setups and needs to be individually determined. Periodical calibrations are necessary to account for contamination and de-adjustments.

## 2   Current methods for path length determination

The determination of the mirror reflectivity $R$ by measuring the mirror transmission $\Theta$ within the instrument itself is not viable, as besides $\Theta$ also the absorption $A$ has to be measured: $R = 1 - \Theta - A$. Additionally, the mirror properties may vary across

the mirror surface (e.g., due to local contamination) and need to be characterised under illumination of the actual measurement configuration. Table 1 shows different methods used to determine the path length in resonators. In principle, $L(\lambda)$, and $R(\lambda)$ (via eq. 1), can be determined from either the temporal behaviour of the intensity of a modulated light input or the light intensity variation when absorbers with known optical properties are introduced into the resonator. For the former, $L(\lambda)$ and the decay time $\tau(\lambda)$ of the resonator photon charge are connected by

$$L(\lambda) = \tau(\lambda)c \qquad (2)$$

where $c$ is the speed of light. For commonly established path lengths $L = 1 \dots 10 \, \mathrm{km}$ this corresponds to $\tau = 3 \dots 30 \, \mu\mathrm{s}$ and $\frac{1}{\tau} = 30 \dots 300 \, \mathrm{kHz}$. In the following, we discuss the various methods in detail.





**Table 1.** Strengths (+) and limitations (–) of current methods for the path length determination in optical resonators.

| | Time resolved methods | | | Methods employing absorbers | | |
|---|---|---|---|---|---|---|
| | **BB-CRD** | **NB-CRD** (this study) | **ICOM** (this study) | **Rayleigh-scatterer** | **calibration gases** | **O$_2$–O$_2$ CIA, H$_2$O** |
| **absolute accuracy** | – | + | + | + | + | ○ |
| **spectral resolution** | – | ○ | + | + | ○ | – |
| **consumables** | + | + | + | – | – | + |
| **cost** | ○ | ○ | ○ | ○ | – | + |
| **automation** | + | + | + | ○ | – | + |
| **limitation** | no spectral resolution | limited spectral resolution | | gas supply, lab method | gas supply, lab method | consistency check only |

## 2.1 Time resolved methods

**Broad band cavity ring-down (BB-CRD)** In Cavity ring-down (CRD), the decay of light intensity leaving the cavity is moni-
tored for a modulated light source. In the simplest case, a monochromatic laser is used together with a fast photo detector,
commonly a photo multiplier (O'Keefe and Deacon, 1988). Path length information is only retrieved at the laser wave-
length. When broad-band light sources are used instead (broad band CRD, BB-CRD), a multi-exponential decay in the
signal is observed, as spectrally differently populated path lengths are superposed (Ball and Jones, 2003; Meinen et al.,
2010). The spectral information of $L(\lambda)$ is concealed and cannot be reconstructed without further assumptions.

**Narrow Band Cavity Ring-Down (NB-CRD)** We present here a further development of the CRD method that also deter-
mines spectral information of the mirror reflectance by combining broad light sources with tunable narrow band filters.
NB-CRD is described in detail in Sect. 3.

**Integrated calibration by means of optical modulation (ICOM)** This study establishes ICOM as powerful new time re-
solved method which adds only little complexity. It is described in Sect. 4.

More time resolved calibration methods exist (e.g., Phase shift cavity ring down absorption spectroscopy (Engeln et al., 1996;
Langridge et al., 2008), high frequency clocked 2D-spectrometer (Ball and Jones, 2003)), which are however prohibitively
expensive and not suitable as a mere calibration technique.

## 2.2 Methods employing absorbers

Other than time resolved calibration methods, absorber-using calibration methods exploit the fact that the optical properties
of gas in the measurement cell affect the transmission through the cavity. Operating the light source continuously at a con-
stant intensity, slow dispersive detectors (spectrometers) can be used to retrieve spectral information. The extinction in the





resonator can be adjusted by introducing gases exhibiting suitable absorption and scattering properties (Rayleigh and Mie) or an antireflection-coated optical substrate of known loss (Varma et al., 2009). The challenge is to accurately control cavity extinction properties, i.e., to introduce adequately strong absorbers or scatterers at accurate concentrations. An inherent prob-

lem of all absorber-using calibration techniques is low sensitivity at short path lengths, i.e. when the light attenuation is small. Additionally, intensity fluctuations of the light source can affect the retrieved path length, if the total absorption instead of the differential absorption is used.

The most popular absorber calibration methods are the following:

**Rayleigh scatterer** One can replace the sample gas in the measurement cell with pure gases of known Rayleigh scattering

coefficient, e.g. helium (Washenfelder et al., 2008). This gas is a less effective scatterer than air, such that more light passes to the spectrometer. This method is relatively cheap, but difficult to apply in the field: The handling and supply of gas during field campaigns (in particular aircraft) is critical. The complete purging of the measurement cell can be hard, both to accomplish and verify. This method relies on a stable light source and transfer optics.

**Calibration gases** Absorbing gases imprint their characteristic (differential) absorption signature on the retrieved spectrum.

The spectral shape of the path length can be derived from the relative strength of the individual absorption structures. If the absorber is introduced at a known concentration, even an absolute path length retrieval is possible (Thalman and Volkamer, 2013). However, many relevant calibration gases (e.g. $NO_2$, $SO_2$) are toxic, not stable and require production on site, they might not cover the full wavelength range, or the accuracy of literature cross-sections are insufficient.

**$O_2$–$O_2$ CIA, $H_2O$** $O_2$-$O_2$ collision induced absorption (CIA) occurs correlated to the concentration of oxygen in the entire

UV-Vis spectral range (Finkenzeller and Volkamer, 2022) and can be predicted accurately if the air temperature, pressure, and humidity are known. Such, oxygen is an ubiquitous calibration gas (Platt et al., 2009; Thalman and Volkamer, 2013). However, the density of differential absorption features is insufficient to spectral characterise the path length. Oxygen can only be used as an consistency check. Similarly, the absorption due to water can be compared to parallel measurements to check for consistency.

While all above methods work in principle, they still pose a substantial complication in the application of optical resonators and do not combine high spectral and absolute accuracy. Here, we present two new methods, (1) Integrated Calibration by means of Optical Modulation (ICOM), which yields high spectral and absolute accuracy in a relatively simple setup, easing the efforts needed in calibration up to now. (2) We also present a closer look at Narrow Band Cavity Ring-Down (NB-CRD).

# 3 Narrow Band Cavity Ring-Down

## 3.1 Method description

The extension of cavity ring-down that allows to retrieve wavelength dependent path length information is to place a wavelength discriminating element in the instruments light path. Ideally, the transmitted wavelength window should be tunable over the





full range of wavelengths. One convenient, simple, and robust solution is to exploit the sensitivity of interference band-pass filters to the light incidence angle. The higher the incidence angle, the shorter the centre wavelength of the transmitted light (Fig. 1, Pollack (1966)). At the same time, the transmission peak is slightly broadened and the peak transmission is reduced. The maximum tilt angle is determined geometrically, when the effective section of the filter (perpendicular to the incident light) becomes smaller than the beam diameter. Typically, the usable wavelength range is on the order of 30 nm or 10 % of the centre wavelength. This approximately matches typical bandwidths of high-reflectivity mirrors and allows spectrally complete path length information for optimal mirror-filter combinations. The spectral tuning range could be extended by using multiple interference filters.

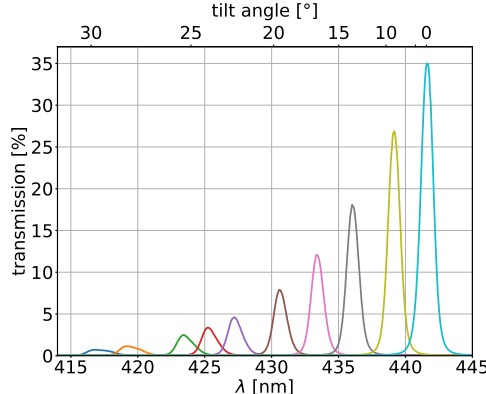

**Figure 1.** Tilt angle sensitivity of transmission characteristics of interference band-pass filter with 441.6 nm nominal centre wavelength. Measured transmission curves are shown for tilt angles from $0°$ (orthogonal, 441.6 nm centre wavelength) to $30°$.

## 3.2 Instrument implementation

Fig. 2 shows a schematic of the instrumental setup including a NB-CRD system. To record ring-down curves, a mirror is introduced into the light path, directing the light exiting the resonator onto a photomultiplier tube (PMT, *Hamamatsu H6780-01*). The PMT signal is amplified by a custom three-stage amplifier and analysed by a small USB oscilloscope (*PicoScope 2204*). The highest sensitivity of the utilised PMT is at 400 nm. A *Schott BG25* band pass filter directly in front of the PMT blocks light with wavelengths longer than 470 nm. The interference filter used in this study (*LOT Oriel 442FS02-50*) has a nominal centre wavelength of 441.6 nm at a nominal FWHM of 1.0 nm. The LED feeding the resonator can be switched from the usual constant current operation into a pulsed mode using custom circuitry, in which it is turned on and off with a frequency of 8.7 kHz. A LabVIEW interface is used to control the tilt angle of the filter, record ring-down signal from the oscilloscope, average data, fit the model function and save the results.




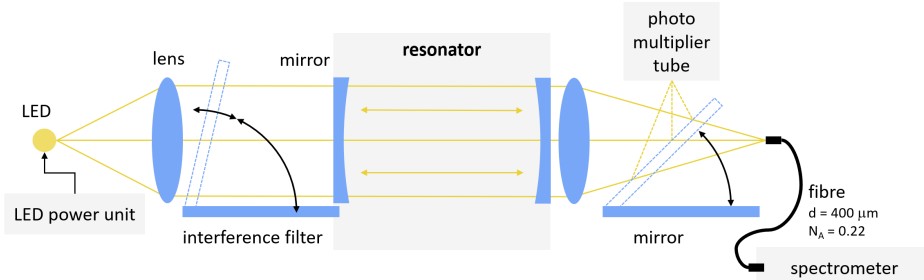

**Figure 2.** Schematic of the CE-DOAS instrument assembly fitted for NB-CRD calibration. For spectrally resolved ring-down measurements, the interference filter can be driven into the light path under different angles. A mirror is used to deflect the beam into a photomultiplier tube.

### 3.3 Calibration routine

Path length calibration using NB-CRD is done as follows. First, by acquiring spectra under constant illumination and different filter angles, the spectrometer wavelength calibration is used to determine the angle-specific centre transmission wavelength. Then, the LED is switched to pulsed operation and, for each selected filter angle, i.e. wavelength, ring-down signals are recorded with the PMT, of which 1000 are averaged. The path length is determined from an exponential fit to the ring-down signal. Usually, three sets are taken for each position to determine their standard deviation as measure of precision. The wavelength-resolved calibration curve is derived by fitting a polynomial to wavelength mapped data points. Depending on the number of sampling points and ring-down measurement sets per point, a calibration $L(\lambda)$ takes between 30 and 60 minutes.

### 4 ICOM

The temporal behaviour of systems is commonly sampled with detectors that allow exposure times $T_{\mathrm{E}}$ significantly shorter than the characteristic setup time scales $\tau$. Spectrometers used in CE-DOAS applications commonly have a minimum exposure times of a few ms, much longer than typical decay times of few $\mu$s. In ICOM, the time resolution of the spectrometer is effectively enhanced to the time resolution of an optical modulator added between the resonator and the spectrometer.

Fig. 4 illustrates the microscopic principle of ICOM, i.e., the evolution of physical parameters during individual cycles. The light source is operated in a modulated mode (pulsed) with period $T$ and delivers an intensity $I(t)$ (Fig. 4A). The photon population in the resonator changes correspondingly to the light input; the temporal evolution of the population, i.e., the shape of rise and decay, is determined by the decay time $\tau(\lambda)$. Photons are lost from the resonator either by scattering, absorption, or by transmission through the mirrors (Fig. 4B). The transmission $\theta(t)$ of the optical modulator between the resonator and the spectrometer oscillates between blocking and transmission of the light, with the same frequency $\nu$ as the light source and with a distinct, adjustable phase relation $\phi$ to the light input signal (Fig. 4C). The intensity reaching the spectrometer is the product of unblocked output (proportional to photon population in the resonator) and transmission. The spectrometer does not resolve





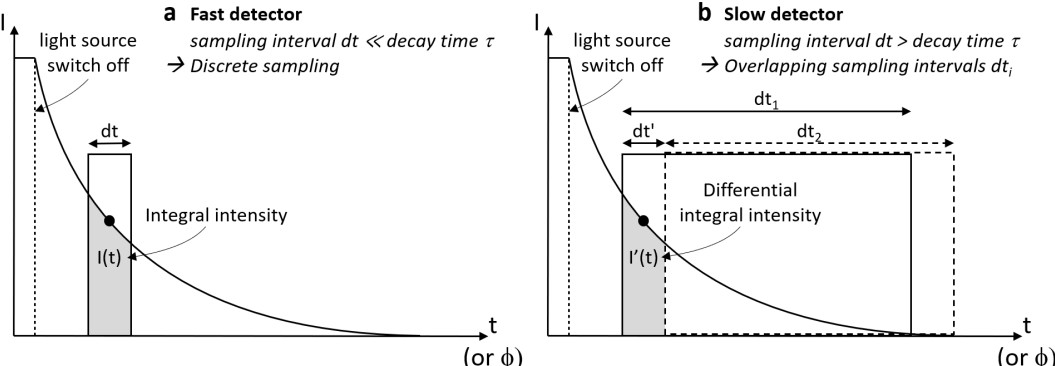

**Figure 3.** Sampling of temporal evolution using fast and slow detectors. In conventional sampling (panel a), a fast detector with sampling interval shorter than the characteristic time constant allows to sample the temporal evolution point by point. If slow detectors are used (panel b), the same information can be retrieved by comparing overlapping sampling intervals.

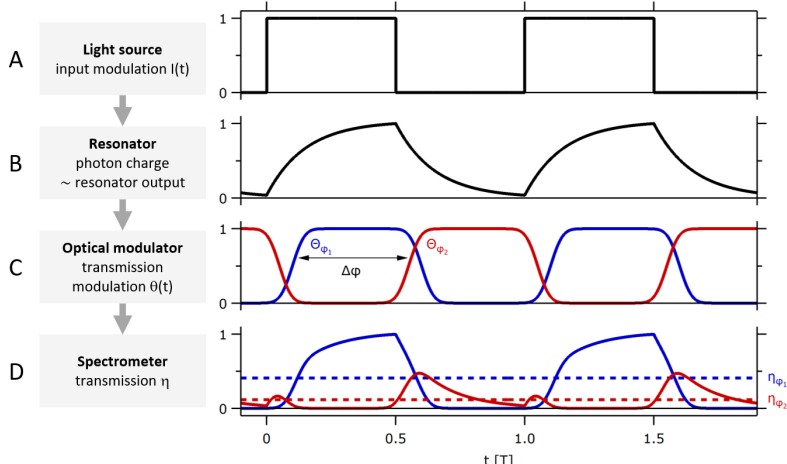

**Figure 4.** Temporal evolution of physical properties in apparatus within two exemplary pulse cycles: A: Light input $I(t)$ from the light source. B: Photon population in the resonator respectively output towards the spectrometer. C: Transmission $\theta(t)$ of the optical modulator for two exemplary phases. D: instantaneous (solid) and apparent (dashed) transmission $\eta$ to the detector. The apparent transmission $\eta$ is a function of the phase $\phi$, $\tau$, $I(t)$, $\theta(t)$ and the modulation frequency $\nu$.

the instantaneous intensity but rather integrates the arriving photons over many pulse cycles (dashed lines in Fig. 4D). We refer to the averaged ratio of light intensity reaching the modulator and leaving the modulator (being transmitted) as apparent





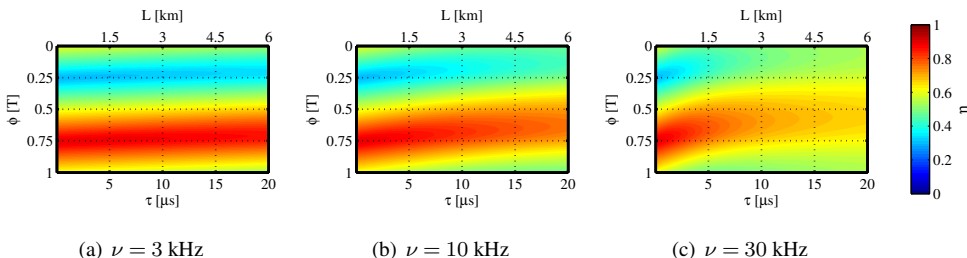

(a) $\nu = 3$ kHz  (b) $\nu = 10$ kHz  (c) $\nu = 30$ kHz

**Figure 5.** Apparent transmission $\eta(\tau, \phi) = \frac{\text{mean intensity at spectrometer}}{\text{mean intensity at modulator}}$ (colour coded) of the demonstration setup (Sect. 5.1, quasi rectangular $I(t)$- and $\Theta(t)$-profile), modelled for three different modulation frequencies $\nu$. The characteristic landscape of $\eta(\tau, \phi)$ can be used to retrieve the path length.

transmission $\eta$:

$$\eta = \frac{I_{\text{modulator}}^{\text{reaching}}}{I_{\text{modulator}}^{\text{leaving}}} = 1 - \frac{I_{\text{modulator}}^{\text{blocked at}}}{I_{\text{modulator}}^{\text{leaving}}} \tag{3}$$

$$= \eta(\lambda, \tau(\lambda), \phi, \nu, I(t), \theta(t)) \in [0, 1] \tag{4}$$

Critically for the ICOM idea, $\eta$ is a function of decay time $\tau$, i.e., path length $L$. It also depends on the phase $\phi$, modulation frequency $\nu$, and the direct and indirect modulation profile $I$ and $\Theta$. Using the modulation profiles $I$ and $\Theta$ of the prototype employed in this study (Sect. 5.1), $\eta$ is given in Fig. 5. Notably, even when the light source and the modulator are operated at a modulation frequency $\nu$ considerably slower than the corresponding decay times $\tau$ (approximately one order of magnitude), sufficient sensitivities for for the determination of the path length can be achieved. This is clear in Figure 3, as the intensity is still retrieved from a differential measurement. Even for modulators with non-rectangular transmission profiles the temporal behaviour can be exploited, as long as the transmission profiles are characterised.

## 4.1 Shutter ratio and light pulse width

Both the pattern of the direct modulation $I(t)$ (of the light source) and the indirect modulation $\theta(t)$ (of the optical modulator) can be adapted to maximise the sensitivity for the retrieval of the path length. If the shutter transmitted only during a very narrow, i.e. short time window and else blocked the light (mean transmission $\bar{\theta} \to 0$, $\delta(t)$-like transmittance), the exponential decay could be well explored point by point, without any broadening of the curve. However, at the same time, very little light would be transmitted to the spectrometer, resulting in a poor signal-to-noise ratio (or long measurement times). In the other extreme case, light throughput could be maximised with continuous transmission ($\bar{\theta} \to 1$), but all temporal information would be lost. Our simulations suggest that — depending on the expected path length range — best sensitivities $S$ (Sect. 4.3) are achieved for a shutter ratio $\bar{\theta} \sim 30\%$ (i.e. 30% mean transmission for a quasi-rectangular temporal transmission profile $\theta(t)$) and a rectangular light pulse width of $\sim 50\%$. The data presented in this study base on the used instrumental setup





(Sect. 5.1), where a slit wheel as modulator with $50\%$ transmittance and a LED with a $50\%$ duty cycle was applied. This way,
approximately $80\%$ of the potential sensitivity of ICOM towards the determination of $L(\lambda)$ is exploited.

## 4.2 Operation mode: Single shot mode and phase sweep mode

There are two modes in ICOM to determine the path length which are depicted in figure 6.

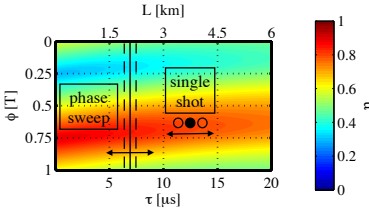

**Figure 6.** Illustration of the two ICOM operation modes:

Phase sweep mode: The apparent transmission $\eta$ (eq. 4) is measured at a quasi continuous set of phases. The acquired $\eta^{\mathrm{Meas}}(\phi)$ curve is then fitted into the modelled $\eta^{\mathrm{Mod}}(\phi, \tau)$ landscape to retrieve the path length.

Single shot mode: The apparent transmission $\eta$ is sampled at a fixed phase $\phi_0$. The acquired $\eta(\phi_0)$ is then fitted to a modelled $\eta(\phi_0, \tau)$ curve to retrieve the path length.

(1) The simplest approach to determine $\tau(\lambda)$ would be to set the phase relation $\phi$ between $I(t)$ and $\theta(t)$ to a fix value $\phi_0$, where $S'(\phi, \tau) = \frac{\frac{\mathrm{d}\eta}{\eta}}{\frac{\mathrm{d}\tau}{\tau}}$ is high (eq. 5, figure 5, single shot mode). $\eta(\phi_0)$ could be determined as ratio of the intensity measured

once with an operational optical modulator and once with a permanently transmitting modulator. $\tau$ can be determined as $\eta(\phi_0)$ is a function thereof. This approach is relies on the comparison of two intensities and is therefor susceptible to drifts in light input intensity and transmission of transfer optics, and offsets of $\phi_0$.

(2) A more robust interpretation of $\eta$ is readily possible if the phase $\phi \in [0, T]$ is swept quasi-continuously (phase sweep mode). This way, a $\eta(\phi)$-spectrum is acquired, whose characteristic shape can be used for the path length retrieval. The absolute

intensity is not relevant. The variation in the $\eta(\phi)$-spectra for different decay times is visualised in figure 7 for different modulation frequencies. In evaluation, measured transmission spectra $\eta^{\mathrm{Meas}}(\phi)$ are compared to modelled transmission spectra $\eta^{\mathrm{Mod}}(\tau, \phi)$.

## 4.3 Theoretical precision of the idealised ICOM setup

The precision of the phase modulation ICOM approach is determined by how precise the shape of $\eta(\phi)$ can be measured and

how much the shape of $\eta(\phi, \tau)$ varies with changes in $\tau$. For a specific phase $\phi$, we define the single shot sensitivity $S'$ as the relative change of the transmission $\eta$ for small relative changes in $\tau$:

$$S' = \frac{\frac{\mathrm{d}\eta}{\eta}}{\frac{\mathrm{d}\tau}{\tau}} = S'(\tau, \phi) \tag{5}$$





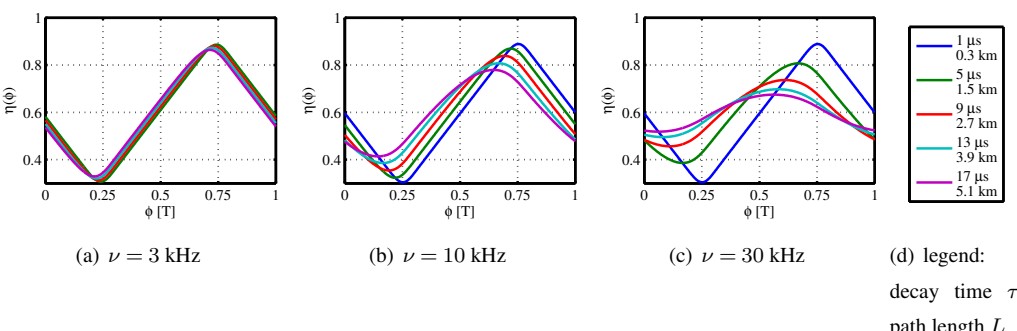

**Figure 7.** Apparent transmission $\eta_{\tau_i}(\phi)$, as defined in eq. 4, of the demonstration setup (Sect. 5.1, quasi rectangular $I(t)$- and $\Theta(t)$-profile), for three different modulation frequencies $\nu$. Both the variable shape and the position of the extrema can be exploited for the retrieval of $L(\lambda)$. Sufficient modulation frequencies $\nu$ are necessary for reasonable sensitivity.

Its dependency on $\phi$ and $\tau$ is visualised in figure 8. In phase modulation mode, a set of $n$ transmissions at different phases $\phi_i$

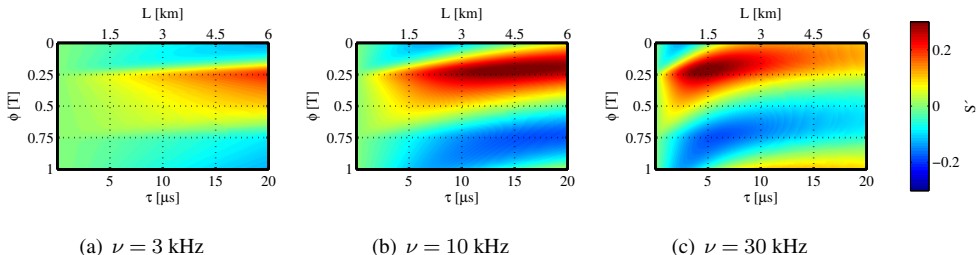

**Figure 8.** Single shot sensitivity $S'(\phi, \tau) = \frac{\frac{d\eta}{\eta}}{\frac{d\tau}{\tau}}$ (colour coded) of the demonstration setup (Sect. 5.1, quasi rectangular $I(t)$- and $\Theta(t)$-profile), for three different modulation frequencies $\nu$. Fields of high $|S'|$ depend indicate where ICOM is sensitive towards retrieving $L$.

is acquired: $\{\eta_i(\tau, \phi_i)\}_{i=1...n} =: \{\eta_i\}_{i=1...n}$. For the analysis of this case we define the sensitivity $S$ for relative changes in $\tau$:

$$S = \sigma\left(\{S'_i\}_{i=1...n}\right) = \frac{\sigma\left(\left\{\frac{d\eta_i}{\eta_i}\right\}_{i=1...n}\right)}{\left(\frac{d\tau}{\tau}\right)} \quad (6)$$

where $\sigma$ denotes the standard deviation. The standard deviation mathematically reduces information from the set of individual sampling points into one scalar and is (amongst possible others) a measure for how much the transmission curve changes with changing $\tau$. One feature of the standard deviation exploited here is that a uniform scaling between two curves (e.g. two curves differ only by a constant scaling factor as a result of intensity drift in light source intensity) does not affect $S$, whereas a change

in shape of $\{\eta_i\}_{i=1...n}$ (i.e. there is a non-trivial transformation between two curves) does contribute to $S$.

It turns out that for our demonstration setup with rectangular light profile $I(t)$ and quasi-rectangular transmission profile $\Theta(t)$, a modulation frequency $\nu = 10\,\text{kHz}$ yields a sensitivity $S \geq 0.1$ for decay times larger $5\,\mu\text{s}$ (figure 9). Faster modulation





frequencies allow higher sensitivities at short path lengths and vice versa. Starting from $S \geq 0.1$, we can estimate the precision

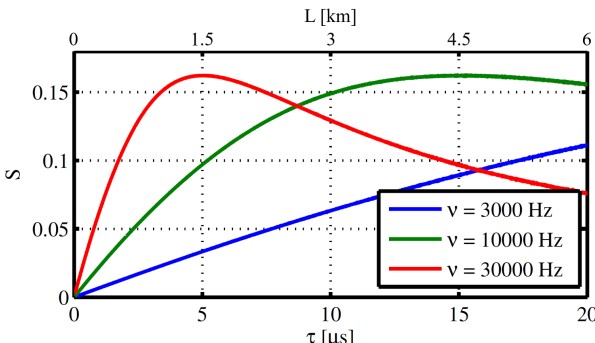

**Figure 9.** Sensitivity $S(\tau)$ (eq. 6) of the demonstration setup (Sect. 5.1, quasi rectangular $I(t)$- and $\Theta(t)$-profile) for three different modulation frequencies $\nu$. At $\nu = 3$kHz, $S$ increases nearly linearly with decay time in the investigated path length range. At higher $\nu$ there are optimal sensitivities at certain decay times. $\nu$ should to be chosen accordingly to achieve maximum sensitivity $S$ in the expected path length ranges.

in $L$ in measurements, not taking into account systematic errors however. Let the precision of the individual transmission

(intensity) measurement at the detector (e.g. one spectrometer channel) be a typical value of about:

$$\sigma \left( \left\{ \frac{\mathrm{d}\eta_i}{\eta_i} \right\}_{i=1...n} \right) \approx 10^{-3} \tag{7}$$

Then, the assumption (7) and $S \geq 0.1$ in equation (6) yield the relative statistical uncertainty in path length:

$$\frac{\Delta \tau}{\tau} = \frac{\Delta L}{L} \leq \frac{10^{-3}}{0.1} = 10^{-2} \tag{8}$$

The phase space does not need to be sampled with constant increments of $\phi$. Selected sampling of the transmission curve $\eta(\tau, \phi)$

at phases of high information content could enhance the precision of the method. However, in this study only equidistantly distributed phase series were considered.

## 5 Demonstration of ICOM

In the following we discuss the practical realisation of an instrument based on the theoretical considerations and model calculations presented above.

## 5.1 Demonstration setup

The schematic description of the resonator and components used in the demonstration setup is given in figure 10. The optical modulator *Thorlabs MC2000* with 100-slot disc *Thorlabs MC1F100* (transmission ratio 50%) was used. Its electronic control unit allows the digital selection of modulation frequencies $\nu$ of up to 10 kHz. Its TTL trigger output with selective delayed phase





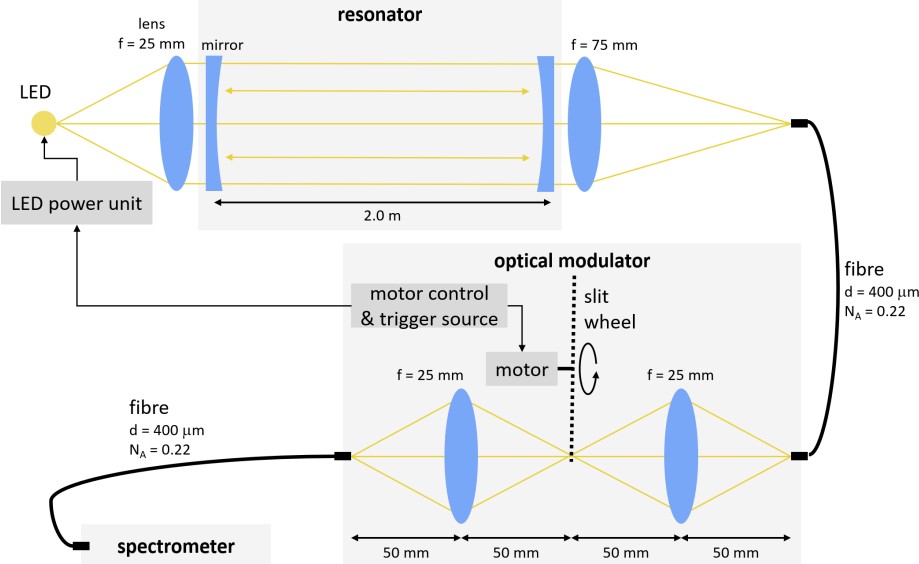

**Figure 10.** Scheme of the experimental setup used in this study. The optical modulator is introduced in between the resonator (bottom) and the spectrometer.

$\phi \in \{0°, 1°, \ldots, 360°\}$ feeds the power supply unit of the light source (LED, Anthofer (2014)), generating a quasi rectangular
light pulse profile with 50% duty cycle.

In the standard CE-DOAS instrument a fibre of $400\,\mu m$ diameter and a numerical aperture $N_A = 0.22$ is used to connect the resonator exit and the spectrometer assembly. This single fibre was now split up into two fibres, with the modulation optics in between. The exit of the first fibre is projected onto the plane of the slit wheel by a 1" plano-convex lens of 25 mm focal length. Object distance and image distance are 50 mm each, resulting in a unity magnification. After passing the chopper wheel, the
image of the fibre is then projected by a second lens onto the fibre connected to the spectrometer.

An unfortunate artefact of the demonstration prototype is due to the fact that the used electronic control unit has two distinct operation regimes (phase $\phi \in [-90° \ldots 90°]$ and $\phi \in (90° \ldots 270°))$, i.e., at the transition a discontinuity in measured intensities appears. As a consequence, special care had to be taken to connect both regimes of $\phi$.

### 5.2  Modelling of apparent transmission $\eta$

The apparent transmission for different $\tau$ and $\phi$ is modelled numerically, using laboratory-determined light pulse profile $I(t)$ of the LED and the measured transmission profile $\theta(t)$ of the optical modulator. The respective profiles were recorded with a PIN photo diode and a transimpedance amplifier (Anthofer, 2014). The spacing between neighbouring $\phi$ was 1°, and 30 ns for $\tau$ (equivalent to 9 m), comparable to measurement noise. Such, a look-up table of $k\,\eta_{\tau_k}^{\mathrm{Mod}}(\phi)$ spectra ($\phi \in \{0°, 1°, \ldots, 360°\}$) is generated for later comparison to measured data.





### 5.3 Data acquisition

Data were acquired with the software *DOASIS* (Kraus, 2006). It communicates with the spectrometer (*Avantes AvaSpec ULS2048L-U2*) and controls the phase $\phi$. The modulation frequency $\nu$ of the optical modulator, and the LED, respectively, was set to a constant value for a calibration run. Such, sets of measured spectra $\eta_{\phi_i}^{\text{Meas}}(\lambda)$ where $\phi_i \in \{0°, 1°, \ldots, 360°\}$ were recorded. At typical integration times of 6 s, data acquisition requires approximately 40 min.

### 5.4 Data evaluation

The wavelengths $\lambda_j$ of the individual spectrometer channels are evaluated separately. First, the data from the $\eta_{\phi_i}^{\text{Meas}}(\lambda)$ spectra are reordered channel-wise to yield phase spectra $\eta_{\lambda_j}^{\text{Meas}}(\phi)$. Then, each measured phase spectrum $\eta_{\lambda_j}^{\text{Meas}}(\phi)$ is compared to modelled phase spectra $\eta_{\tau_k}^{\text{Mod}}(\phi)$ of the previously generated look-up-table using the Levenberg-Marquardt least $\chi^2$ fitting algorithm:

$$\eta_{\tau_k}^{\text{Mod}'} = A + B \cdot \eta_{\tau_k}^{\text{Mod}}(\phi + \phi_0) \rightarrow \eta_{\lambda_j}^{\text{Meas}}(\phi) \tag{9}$$

Here, $A$ and $B$ are fit coefficients for offset and scaling in intensity (to account for pixel dependent offset, dark current, offset in shutter profile and offset in LED pulse profile); the constant phase offset $\phi_0$ is an instrument parameter. It can either be determined in auxiliary measurements or at wavelengths of known negligible path length. The root mean square (RMS) of the fit residual is taken as measure for the quality of the match. The $\eta_{\tau_k}^{\text{Mod}}(\phi)$ phase spectrum with smallest RMS identifies the most probable decay time, i.e., path length.

To account for the two distinct phase regimes of the used specific chopper control unit (discontinuity in the $\eta_{\lambda_j}^{\text{Meas}}(\phi)$ spectra at $\phi = 90° \leftrightarrow 91°$ & $270° \leftrightarrow 271°$ occurring in the used instrumental setup), the fitting scenario had to be extended by a box reference that allows different offsets in the two regimes:

$$\eta_{\tau_k}^{\text{Mod}'} = A + B \cdot \eta_{\tau_k}^{\text{Mod}}(\phi + \phi_0) + C \cdot \begin{cases} 1 & \phi \in [-90°, 90°] \\ 0 & \phi \in (90°, 270°) \end{cases} \rightarrow \eta_{\lambda_j}^{\text{Meas}}(\phi) \tag{10}$$

Here, $C$ is an additional fit coefficients accounting for the discontinuity.

## 6 Discussion

### 6.1 Accuracy estimation: Comparison of ICOM to NB-CRD & helium calibration

Figure 11 shows the comparison of retrieved wavelengths for an optical cavity in the blue wavelength range using ICOM, helium, and NB-CRD. The wavelength calibration of the spectrometer was performed using krypton emission lines.

**Helium calibration** The wavelength calibration of the spectrometer was performed using krypton emission lines. The measurement cell of the instrument was then purged with ambient air and helium in repeatedly to verify complete flushing





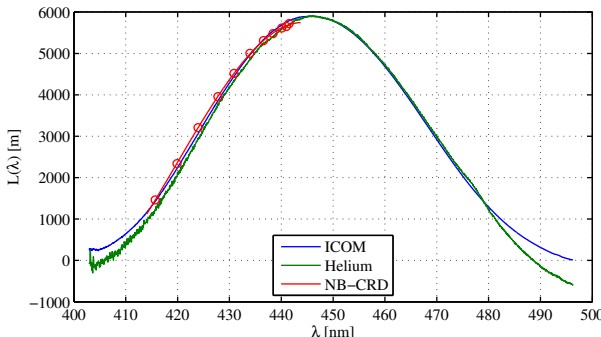

**Figure 11.** Comparison of spectral path lengths retrieved with ICOM (blue line), helium, (green line), and NB-CRD (red circles and line). The NB-CRD information is limited to $\lambda \leq 442$ nm, corresponding to the band pass filter tuning range; the circles indicate the sampled wavelengths that were interpolated by a second order polynomial. Significant differences only exist at the spectral margins, were the helium calibration determines non-physical negative path lengths.

from reproducible intensity changes. Following the evaluation according to Washenfelder et al. (2008), the different Rayleigh absorption between air and helium was used to derive the light path. The literature Rayleigh scattering cross-sections for helium and a standard air mixture were calculated accordingly to Bodhaine et al. (1999) (at $\lambda = 430$ nm: $\sigma_{\text{He}} = 1.93 \cdot 10^{-28}$ cm$^2$ molec$^{-1}$, $\sigma_{\text{air}} = 1.24 \cdot 10^{-26}$ cm$^2$ molec$^{-1}$); for the absorption cross section of O$_4$ data from Thalman and Volkamer (2013) were used (peak absorption at $\lambda = 446$ nm: $\sigma_{\text{O}_4} = 5.53 \cdot 10^{-47}$ cm$^5$ molec$^{-2}$). The resulting path length curve is included in figure 11.

Incomplete flushing of the resonator volume with helium can bias the path length short. Additionally, the exchange of the resonator gas-bath by helium can alter the instrument illumination, compared to the illumination during measurements. This can bias the retrieved path length long or short (compare non-physical negative path lengths at 405 nm and 495 nm wavelength, Fig. 6.1). While the helium calibration achieves small statistical uncertainties of few 10 m (same order as noise on the curve), systematical errors due to incomplete flushing and drift in the light source intensity are likely much larger.

**Narrow Band CRD calibration** A NB-CRD calibration following Sect. 3 was performed for comparison. With the available filter, a path length retrieval for wavelengths longer than 442 nm was not possible. The ring-down signal was recorded at eleven different tilt angles. The individual $L(\lambda_i)$ were fitted by a second order polynomial. $L(\lambda)$ and the $L(\lambda_i)$ are included in figure 11.

**ICOM** At $\nu = 10$ kHz, for each phase $\phi \in [0°, 1°, \ldots, 360°]$ the transmission to the spectrometer was stored in spectra with 6 s integration time. Data acquisition took 40 min per run. The path length retrieval was performed according to Sect. 5. The average path length from five consecutive runs is included in figure 11.





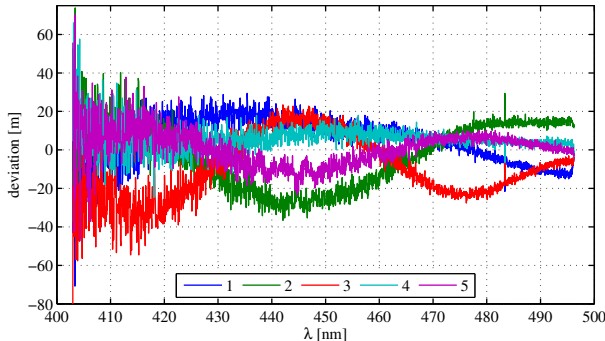

**Figure 12.** Demonstration of ICOM precision: Deviation $\Delta L_i(\lambda) = L_i(\lambda) - \overline{L(\lambda)}$ of five individually acquired $L_i$ at $\nu = 10\,\text{kHz}$ from the mean path length $\overline{L(\lambda)} = \sum_{i=1}^{5} L_i(\lambda)/5$. The variable noise in the spectrum is due to the spectrally non-homogeneous light intensity distribution. The path length peaks at 443 nm ($\sim 6\,\text{km}$, figure 11) and is essentially zero at the margins of the spectrometer ran.

The differently acquired $L(\lambda)$ (figure 11) show a good agreement for path lengths above few km with differences smaller $\sim 100\,\text{m}$. Strikingly, the path length retrieved with helium calibration seems to be wavelength-shifted against the others. This could be due to inaccurate scattering cross-sections used in the helium calibration evaluation. It is difficult to specify the absolute accuracy of ICOM, as the comparison of both the NB-CRD and helium calibration apparently do not exceed the accuracy of ICOM.

### 6.2 Precision estimation: Repeated measurements

The precision of ICOM was assessed by five successive measurements of the path length of the same optical resonator, as in Sect. 6.1. The deviations of each path length from the mean path length are illustrated in figure 12. The five retrieved ICOM path lengths are very similar. The statistical uncertainty of the path length can be estimated from the channel-to-channel noise to be on the order of few 10 m for individual runs. A possible explanation for the systematical deviations of several 10 m is a slightly shifting alignment of the optics, resulting in a shifting phase offset $\phi_0$. Electronic optical modulators would not be sensitive towards such an effect. However, the reproducibility seems to be better than 50 m, independent of absolute path length.

### 6.3 Agreement of modelled and measured data

Figure 13 shows the agreement between measured and modelled phase spectra from a single run at $\nu = 10\,\text{kHz}$, consisting of statistical noise and systematic structures. The low noise, compared to the magnitude of systematic residual structures, demonstrates the high brightness of the ICOM approach, allowing swift and precise calibrations. The most relevant instrument interference in the demonstration prototype stems from the two distinct operation regimes of the phase control unit, creating discontinuities at $\phi = 90°$ and $270°$. These are satisfactorily but not fully accounted for by the box reference included in the fit algorithm (Sect. 5.4), and should be no concern in future setups. We hypothesise that the residuals may partially due to jitter





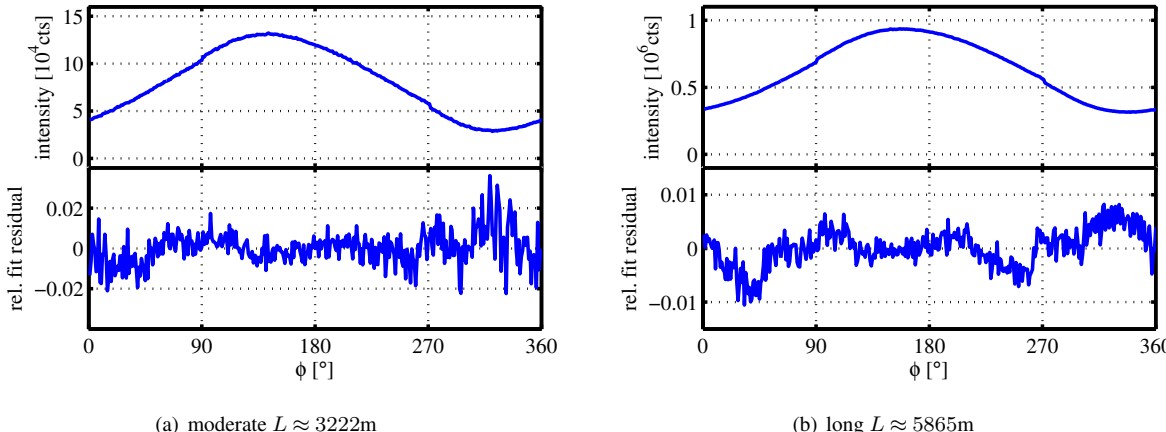

(a) moderate $L \approx 3222\,\text{m}$                                    (b) long $L \approx 5865\,\text{m}$

**Figure 13.** Acquired intensity as function of phase $\phi$ and corresponding relative fit residual $\eta^{\text{Meas}}/\eta^{\text{Mod}'} - 1$ for modulation frequency $\nu = 10\,\text{kHz}$. The congruence of modelled transmission and measured transmission is in the low percent range. The intensity spectra exhibit a discontinuity at $\phi = 90°$ and $270°$ due to an instrumental artefact.

of the mechanical optical modulator, i.e., deviations from the average modulation $\theta(t)$ that lead to a distortion of modified apparent transmission $\eta$, and slight imperfections in the characterisation of $I(t)$ and $\theta(t)$.

### 6.4 Required modulation frequencies

To investigate the minimum modulation frequency necessary for an accurate path length determination in our setup with a
maximum path length of 6 km, the optical modulator was operated at reduced frequencies $\nu = 8, 4, 2, 1\,\text{kHz}$. Figure 14 shows that the respective path lengths agree well down to $\nu \approx 2\,\text{kHz}$, where systematical differences start to appear. The path length for $\nu = 1\,\text{kHz}$ is non-physically coarsely step-shaped, suggesting numerical problems. However, ICOM appears to be applicable also for modulation frequencies in the low kHz range that are accessible by slower modulators.

### 6.5 Applicable optical modulators

ICOM requires optical modulators that operate at frequencies of several kHz to ensure sensitivity, and that have an aperture comparable to the beam diameters (e.g., the fibre diameter $400\,\mu\text{m}$). A high contrast, i.e., a strong modulation, is desirable. A potential wavelength dependence of the modulation would need to be characterised and considered appropriately. Table 2 lists various types of devices for the indirect modulation of optical beams. For the proof of concept, a rotating slit wheel was used because of its easy implementation and high performance. Slit wheels exploiting the Moire effect, where instead of a slit
a grating is swept over by a complementary grating, allow even higher frequencies of several 10 kHz. Tuning fork choppers are extremely robust but limited by the small beam diameter they permit at high frequencies. Liquid crystal optical modulators are promising, as they avoid moving parts and are likely easy to integrate into optical setups. Their maximum modulation



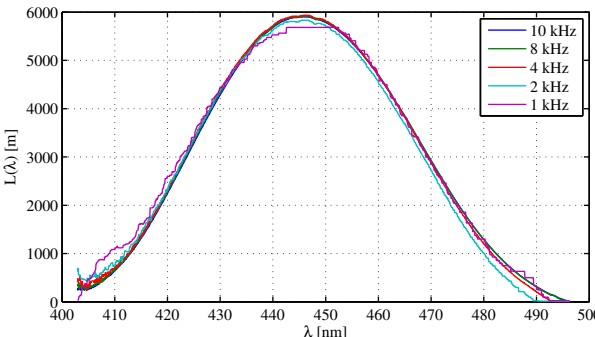

**Figure 14.** Successful ICOM path length retrieval using artificially reduced modulation frequencies $\nu$.

**Table 2.** Available optical modulators. ICOM demands modulation frequencies in the kHz range for fibre-sized beam diameters.

| Method | Principle | $\nu_{\mathbf{max}}$ | Comment |
|---|---|---|---|
| digital light processing (DLP) | bistable micro-electro-mechanical mirrors | $< 1\,\mathrm{kHz}$ | used in video projectors |
| liquid crystal | electronically switching polarizer | few kHz | transmission $\theta \in [0, 0.5]$, easy integration in optical setup |
| tuning fork chopper | vanes attached to tines | $< 6\,\mathrm{kHz}$ | robust, $\nu$ fixed, small beam diameter at high $\nu$ |
| slit wheel | moving grating | $< 10\,\mathrm{kHz}$ | used in this study |
| slit wheel /w Moire effect | grating moving relative to complementary grating | $< 100\,\mathrm{kHz}$ | very fast mechanical solution, transmission $\theta \in [0, 0.5]$ |
| acoustic optic modulator | diffraction at sound waves within crystal | some MHz | costly, confined to lasers |
| electro optic modulator | manipulation of refractive index in crystal | some GHz | costly, confined to lasers |

frequency of liquid crystal optical modulators is limited by the autonomous aligning of the crystals after removal of the electric field. Micro-electro-mechanical mirrors (digital light processing (DLP)), acousto- and electro-optic modulators are less relevant

because of their complexity and cost.

Even without an additional optical modulator, it is in principle possible to sample the temporal evolution of a resonator charge with comparably slow detectors (conventional spectrometers). Rather than continuously modulating the light input and measuring mean intensities at the detector, single modulations can be sampled as long as the shift between input modulation and acquisition can be controlled (Fig. 3). The challenge of this approach consists in appropriate timing control and signal-to-

noise ratio, especially when the processing of a spectrum (detector clearing, exposure, read-out) for later software-averaging



takes much longer than the resonator decay time $\tau$. The benefit is its simplicity in that it does not require calibration specific transfer optics.

## 7 Conclusions

The agreement of ICOM, helium, and NB-CRD calibrations demonstrates the feasability of the two new methods. NB-CRD
is a relatively simple method that requires few components (filter, servo, photomultiplier, oscilloscope) and yields robust path length measurements independent of consumables. With servo actuated mirrors and filters, calibrations can be automated. The main challenge consists in achieving full spectral coverage by using band pass filters matched to the spectral range. In ICOM, the path length in the resonator was determined with excellent statistical uncertainty of $\sim 10\,\mathrm{m}$ at the spectral resolution of the spectrometer, independently of the absolute path length. For typical multi-km path lengths this figure translates into a relative
error in $L$ well below 1%. We show that ICOM allows the accurate path length determination also at modulation frequencies of few kHz that are accessible by optical modulators that avoid complexity and consumables. Advantages of ICOM are the high spectral resolution of the spectrometer (as during actual measurements), the high accuracy in path length, and its cost efficient and robust design without consumables and wearables (e.g., if a tuning fork chopper is used, or if single modulations are sampled, Sect. 6.5). ICOM is an integrated calibration, i.e., even during calibration spectral data are acquired that could be
used to retrieve trace gas concentrations. The performance of ICOM can still be enhanced through the optimisation of control parameters. The combination of high performance and easy implementation make ICOM and NB-CRD a promising technique for the spectral path length calibration of optical resonators, where other techniques are not feasible, e.g., in the field or in open-path setups.

*Author contributions.* H.F. developed the idea of ICOM, conducted the experiments, and wrote the manuscript with help of all co-authors.

*Competing interests.* The authors declare no competing interests.



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
