# Peer review of "New methods for the calibration of optical resonators: Integrated Calibration by means of Optical Modulation (ICOM) and Narrow Band Cavity Ring-Down (NB-CRD)"

_Atmospheric Measurement Techniques, 2022_

## Author Response (AR1)

**Response to reviewer comments**

We would like to express our gratitude to the reviewers for taking the time and for providing helpful comments on the manuscript. We address the comments point by point below, with *reviewer comments in black*, responses to comments in blue, and changes to the manuscript in green. Additionally, we carefully re-read the manuscript, corrected typographical errors, and made minor editorial changes.

**Reviewer 1 comments**

*Review of Finkenzeller et al. "New methods for the calibration of optical resonators: Integrated Calibration by means of Optical Modulation (ICOM) and Narrow Band Cavity Ring-Down (NB-CRD)"*

*Finkenzeller et al. provide an innovative discussion of cavity-based calibration techniques. The ICOM technique especially is a new and very different approach to understanding the behavior of light in optical cavities in order to retrieve trace gas concentrations. The paper is of high import to the community, the experiment design is well described and well designed. I recommend publication follow addressing the comments below.*

*Major Comments:*

*While the paper is written from the perspective and application of the CE-DOAS retrieval method, some commentary on how to apply this to the BBCEAS style retrievals using the equations in Fiedler or Washenfelder would be useful for the community. Specifically, how do these calibration methods help to get to the mirror reflectivity (especially for situations where the extinction in the cavity is not know such as open cavity or the mentioned retrieval of data while sampling the path with the ICOM method).*
We appreciate the comment. We have amended the introduction and clarified the connection between path length and mirror reflectivity, with an additional reference to *Fiedler et al 2003.*
Given that $d_0$ and $\varepsilon_\lambda$ are easily determined, the determination of $L(\lambda)$ is therefore closely connected to the determination of the mirror reflectance $R(\lambda)$, which is often used as the characterising parameter and a key parameter in the retrieval of trace gases [e.g., Fiedler et al 2003].

*It would be helpful for the authors to comment on how levels of extinction in the cavity effect the ICOM method. Specifically, how would this be utilized in the cases where the absorbers or extinction severely reduce the effective pathlength in the cavity? At what range of concentrations is the ICOM retrieval valid for all spectra? How do structured absorbers effect the value retrieved?*
Please refer to the response below regarding the separation of different types of extinction during ICOM calibrations. For the interpretation logic that considers the effect of wavelength dependent light path reduction due to, e.g., molecular absorption, see ICAD (*Horbanski et al 2019*) and references therein.

*Line 254: Here it is mentioned that the mirror reflectivity is calculated using Rayleigh scattering of air. Why would air be used instead of N2 which has no CIA features in this wavelength range? (Especially when the oxygen content of synthetic zero air often varies per the manufacturer from 20-24%.)*

Filtered and dried atmospheric air was used in the study as it can be relatively simply provided by passing laboratory air through an appropriate filter (here, charcoal) to remove molecular absorbers. Using this approach, a compressed gas bottle is not needed. The mixing ratio of oxygen of atmospheric air (or well ventilated rooms) is well known, in contrast to synthetic air. Using an atmospheric air mixture instead of a pure Rayleigh scattering gas requires additional terms in the interpretation of calibration data, i.e., $O_2$-$O_2$ CIA and the Rayleigh scattering contributions from the different gases, but these contributions are understood. We have amended the respective paragraph to clarify the used procedure as follows:

The wavelength calibration of the spectrometer was performed using krypton emission lines. The measurement cell of the instrument was then purged with ambient air and helium repeatedly to verify complete flushing from reproducible intensity changes. Following the evaluation according to Washenfelder et al 2008, the different amount of Rayleigh scattering between air and helium was used to derive the light path. The literature Rayleigh scattering cross-sections for helium and air (laboratory air passed through an active charcoal filter and drying cartridge to remove absorbing species and humidity) were calculated accordingly to Thalman et al 2014, Bodhaine et al 1999 (at $\lambda$ = 430 nm: $\sigma_{He}$ = $1.93 \cdot 10^{-28}$ cm$^2$ molec$^{-1}$, $\sigma_{air}$ = $1.24 \cdot 10^{-26}$ cm$^2$ molec$^{-1}$); for the absorption cross section of $O_2$-$O_2$ collision-induced absorption data from Thalman et al 2013 were used (peak absorption at $\lambda$ = 446 nm: $\sigma_{O2O2}$ = $5.53 \cdot 10^{-47}$ cm$^5$ molec$^{-2}$). The resulting path length curve is included in Figure 11.

*How is the state of the mirror to be deconvolved from the combined value of the mirror and other losses in a situation where a calibration gas (free of absorbers of interest) isn't supplied? I would assume that this also may be a concern where the extinction in the cavity (especially for open path operation) undergoes temporal changes.*

ICOM determines the wavelength-dependent pathlength via the total extinction (i.e., the sum of mirror extinction and extinction in the gas) from the comparison of multiple spectra, acquired under different phases. If the extinction due to the gases is known (i.e., concentrations and respective extinction cross sections), the instrument properties (i.e. the mirror extinction) can directly be isolated, compare Eq. 1. In case of additional extinction (of unknown intensity but known shape) due to, e.g., Mie scattering or molecular absorption, this additional extinction (in individual spectra) needs to be determined to allow the determination of the mirror extinction. This simultaneous retrieval of the mirror extinction and trace gas concentrations is in principle possible, as long as the trace gas extinction can be separated from the mirror extinction. This in turn is possible if either the trace gas spectral characteristics are distinct from mirror extinction (i.e., substantial differential extinction but negligible broad-band absorption, e.g., $H_2O$, IO, HCHO, HONO, and $NO_3$, but not $NO_2$ or $I_2$), or if there are periods with negligible concentrations within the set of spectra that are interpreted together. Effectively, the mirror reflectivity (in combination with spectrum-specific trace gas extinction) that explains the set of spectra with smallest residuals is the best guess of the mirror reflectivity. The development of a respective practical algorithm is beyond the scope of this study. If the mirror reflectivity is already known

from previous calibrations and the ICOM procedure is only used to confirm the mirror reflectivity, the concentrations of different absorbers may be determined in a relatively straightforward way for individual spectra.

Given that the light throughput from the light source to the spectrometer is necessarily reduced in the direct modulation of the light source and indirectly in the transmission to the detector, operating the system continuously illuminated achieves a better signal-to-noise ratio and will therefore be the generally preferred measurement mode.

*Are there other practical concerns that need to be addressed such as variability of chip saturation between the full spectrum and the ICOM spectra, as well as how different spectrometers may or may not be suitable for this application? (e.g. how a mechanical shutter operation may vary from using an electronic shutter, or how bleed down the chip for an electronic shutter may or may not effect any of these retrievals).*

The increased brightness when transitioning from ICOM sampling to 'full spectra sampling' should not be critical, if 'full spectra' are analysed relative to 'full reference spectra'. An adequate intensity-linearity is required for the correct interpretation of spectra, especially as the relative differences in intensity during an ICOM scan can be considerable. It is a feature of ICOM that the modulation can be achieved by an additional component that is independent of the spectrometers, i.e., essentially any integration counter can be used. Electronic shutters, e.g., liquid crystal filters, may require a more stringent characterization, potentially exhibiting temperature and wavelength sensitivities. With suitable spectrometers capable of triggering the acquisition of spectra coordinated to the modulation of the light source, it is in principle also possible to record a set of individual modulations which are subsequently software-averaged. In this implementation, ICOM adds little hardware complexity.

*Minor Comments:*

*Line 101: rephrase to "allows retrieval of wavelength dependent path length"*
Rephrased to The extension of cavity ring-down allows the retrieval of wavelength dependent path length information by placing a wavelength discriminating element in the instruments light path.

*Line 245: fit coefficient (remove s).*
Corrected

*Line 251: "ambient air and helium repeatedly" (remove to)*
Corrected

*Line 253: Rayleigh scattering (not absorption). Also, there is no mention of which Rayleigh scattering cross-section is used for Helium.*
Corrected, reference to *Thalman et al 2014* and *Bodhaine 1999* added.

**Reviewer 2 comments**

*Finkenzeller report two new methods for calibrating broadband optical cavities (IBBCEAS/BBCEAS/CE-DOAS). While broadband optical resonators have been used in laboratory and field measurements for over a decade, the question of how to calibrate instruments is still a challenge for researchers. Although several methods have been proposed and are useful in various contexts, they all have disadvantages, whether additional expense or the inconvenience of requiring bottled gases. The paper therefore addresses an important technical aspect of the use of broadband optical resonators.*

*The authors describe the considerations around calibration and propose two methods:*

- *a method for ringdown calibration of narrow wavelength band using narrow bandpass filters. Here, the calibration is carried out across a wide spectral range by exploiting the angle-dependence of the centre wavelength of pass band. This approach is therefore a convenient approach to carry out a conventional ringdown calibration across a wide spectral region.*
- *A more important development is their proposal of Integrated Calibration by means of Optical Modulation (ICOM), which uses both a modulated light source and an additional modulator to characterise the optical cavity pathlength.*

*ICOM in particular is a useful advance that delivers highly accurate measurements of the effective pathlength of the resonator. The authors provide modelled and experimental results of both calibration methods and compare these to a commonly used calibration approach based on differences in Rayleigh scattering of two gases. The results of both new calibration methods are very good and agree well with the Rayleigh calibration, indicating that both methods are accurate and not too expensive.*

*The methods proposed here are likely to be valuable tools for researchers using optical resonators. I have a few comments and questions regarding the paper.*

*Several clarifications would strengthen the manuscript:*

- *The authors should clarify the difference between phase shift measurements (Langridge, 2008) and the approach taken here. They are not the same, but it would be instructive to describe and explain the different approaches more clearly.*
  We have amended the introduction.
  PSCRDS employs a modulated light source (typically direct modulation of an LED, pulse length a few 10 ns) and a fast detector (photomultiplier tube, PMT). Spectral information is gained with a monochromator. The temporal evolution at the selected wavelength is sampled with a phase-sensitive lock-in amplifier. The measured phase shift readily determines the intra-cavity photon lifetime. PSCRDS therefore differs from ICOM in both the hardware and the analysis approach.

- *The text (79-83) should make clear that the Rayleigh scattering approach to calibration requires two pure gases for the calibration, not just one.*
  We agree that the approach relies on the comparison of two gases and have amended the paragraph accordingly. We have also made clear that zero air can in principle be used, if the contribution of $O_2$-$O_2$ CIA is accounted for:
  One can determine the setup properties by comparing gases of known and distinct Rayleigh scattering coefficient, e.g. helium and zero air [Washenfelder et al 2008]. Helium scatters less than air, such that more light passes to the spectrometer. In the case of zero air, $O_2$-$O_2$ collision-induced absorption needs to be considered as well.

- *Use of H2O absorption is challenging for optical cavity measurements because the narrow absorption lines of gas phase H2O are unresolved by typical spectrometers, resulting in apparent non-Beer-Lambert behaviour. This should be clarified in the text.*
  We appreciate the comment and have amended the paragraph accordingly:
  For water, strong absorption in narrow absorption lines that are not resolved by typical spectrometers can lead to apparent non-Beer-Lambert behaviour, and special care may be required in the interpretation [*Langridge et al 2008*].

*The authors put the modulator after their resonator, but they should indicate whether this is necessary, or whether it is possible to put the modulator before the resonator. Presumably, as a product of different time-dependent transmission functions, the two arrangements are mathematically equivalent.*

The optical modulator needs to be introduced into the light path between the resonator and the spectrometer. It cannot be introduced between the light source and the resonator. If introduced directly after the modulated light source, it would effectively only change the modulation of the light source, but no information on the resonator as, in this configuration, external component would be gained. We have further clarified the position of the modulator in the introductory paragraph of ICOM.

*40 min is a long time for the calibration. What is the maximum phase interval that retains sufficient calibration accuracy?*

We agree with the statement that 40 min can be substantial under certain circumstances, e.g., when setting up a new instrument. We agree that the optimization of the data acquisition procedure holds potential to shorten the calibration procedure.

Optimization could include the number of sampled phases, the spacing of phases (uniformly distributed or distributed according to the information density), the spectra acquisition time (both total and individual, to ensure good statistics even when the transmission is low), and sampling sequence of the different phases. Additionally, time may be saved by focusing data acquisition to relevant wavelengths only, i.e., to wavelengths with substantial path length that are later actually used for the retrieval of trace gas concentrations. To systematically explore this parameter space is beyond the scope of this study.

We are reluctant to recommend a maximum phase interval, but are willing to hypothesise based on the number of features that the apparent transmission can be assumed to be composed of. For a typical modulation with a frequency similar to the decay rate, for a variation of the phase from 0° to 360°, the transmission will vary between the four regimes of least-transmitting, ring-up-transmitting, maximally-transmitting, and ring-down-transmitting. Assuming that each of these regimes can be sampled using a small set of phases (e.g., 6 distinct phases), then the characterization of the entire apparent transmission could be achieved with, e.g., 24 phases, or 15° phase separation.

Typical CE-DOAS or ICAD instruments, e.g., ICAD NOx and HONO monitors by Airyx, regularly record reference spectra free $I_0(\lambda)$ of absorbers. The ability to reproduce $I_0(\lambda)$ is evidence for an unchanged mirror state, while a degradation of $I_0$ indicates that a new calibration is needed. Under these circumstances, a calibration may not be indicated for many days or weeks, rendering a calibration duration of ~1 h acceptable. More rapid calibrations, e.g., when setting up a new instrument, can be achieved at the expense of signal-to-noise ratio. The total avoidance of measurement gaps is in principle possible by extending ICOM to include the retrieval of trace gas concentrations during the calibration, albeit requiring an advanced analysis algorithm.

*What is the estimated uncertainty in the Rayleigh method and the NBCRD measurements? Can the authors put a quantitative upper limit on their uncertainty for the ICOM approach (for typical mirror reflectivities)?*

The introduction of different transfer optics (in case of NBCRD) and a different gas (in case of the Rayleigh method) means that a constant optical alignment and instrument illumination between calibrations and measurements is not guaranteed. We interpret that the uncertainty of both the Rayleigh-derived and NBCRD-derived path length is not determined by a lack of precision, but rather limited by systematic uncertainties. We have amended the paragraphs as follows:

Incomplete flushing of the resonator volume with helium can bias the path length short. Additionally, the exchange of the resonator gas-bath by helium can alter the instrument illumination, compared to the illumination during measurements. This can bias the retrieved path length long or short (compare non-physical negative path lengths at 405 nm and 495 nm wavelength, Fig. 11). While the helium calibration achieves small statistical uncertainties of a few 10 m (same order as noise on the curve), systematic errors due to incomplete flushing and drift in the light source intensity are likely much larger. Based on the derived non-physical negative path lengths, the accuracy of the helium-derived path length estimate is on the order of 300 m for the used setup, or 5% of the peak path length.

[...]

The accuracy of the NB-CRD path length estimate is comparable to the accuracy of the helium-derived path length estimate, i.e., on the order of 300 m, or 5% of the peak path length.

We appreciate that the reviewer presses for a number on the uncertainty of ICOM. While ICOM does not necessarily require the modification of the instrument illumination and is therefore likely more accurate, we are unable to provide a number with confidence.

*Fig. 12: It would be more useful to see the percentage uncertainty in the pathlength rather than the absolute value since the pathlength changes enormously over the spectral range.*
We agree that a relative comparison is relevant and could be complementary to the absolute comparison. At the same time, a relative comparison would suffer from division by zero at the spectral margins. We have amended the caption by the percent value for peak pathlength to convey the relative information in addition to the absolute information.

*Minor corrections*

*There are a considerable number of errors in the writing. Some of these are indicated below, but the authors should carefully review and edit their manuscript.*
We have carefully reread the manuscript, corrected typographical errors, and made minor editorial changes.

*16: "along its propagation" is obscure & probably incorrect. I recommend replacing with "along its propagation path" or "as it propagates"*
Changed to "along its propagation path".

*19: rephrase "factor of up to several 10". One to two orders of magnitude?*
Changed to "one to two orders of magnitude".

*23: add "or"*
Changed to "It is a characteristic property of broad-band cavity enhanced absorption spectroscopy (BB-CEAS) or cavity enhanced differential optical absorption spectroscopy (CE-DOAS)"

*31: "commonly reaches A few percent".*
Added.

*32: "determined through sole geometrical considerations." -> "determined solely by geometrical considerations, unlike for multipass cells"*
Changed as suggested.

*40: "Periodic"*
Corrected.

*90: "As such, "*
Changed.

*93: "as a consistency"*
Changed.

*96: "do not combine high spectral and absolute accuracy". It's spectral coverage, not accuracy, that is the issue; "absolute" (of L or R, not wavelength) seems here to mean "very good" (as opposed to "relative accuracy").  I think the authors mean something like "high accuracy with broad spectral coverage".*
We appreciate the concern of the reviewer. Indeed, the combination of sufficient spectral coverage, resolution, and accuracy, i.e., precise and unbiased retrievals for every wavelength, is desired. We have considered the concern and reworded the paragraph for more clarity:
While the above methods work in principle, they substantially complicate the use of optical resonators and are either difficult to implement or do not allow to accurately retrieve the path length with spectral resolution and coverage. Here, we present two new methods, (1) Integrated Calibration by means of Optical Modulation (ICOM), which allows a high accuracy with spectral resolution and coverage in a relatively simple setup, easing the hitherto needed efforts in calibration.

*97: as for 96.*
See response above.

*130-3: Add reference to Fig. 3.*
Added.

*163: change to "were based on the instrumental setup used"*
Changed to "are based on the instrumental setup used".

*171: "This approach relies on"*
Corrected.

*168: "fixed value"*
Corrected.

*Fig 8 caption: Change "depend indicate"*
Corrected.

*223: Start with "A look-up table".  Is k defined previously?*
Sentence structure adjusted to start with "A look-up table"..
In the submitted manuscript, k was introduced in lin 223. We agree that it could be introduced more clearly, which is considered in the revised version of the manuscript, i.e., k is now introduced in the first sentence of the paragraph.

*228: "As such,"*
Corrected.

*251: Sentence unclear.*
Sentence revised to: "The measurement cell of the instrument was then purged with ambient air and helium repeatedly to verify complete flushing from reproducible intensity changes."

*261: Fig 6.1?*
Reference corrected.

*290: Residuals partially what? "Arise"? Unclear*
Corrected.

*Table 1: explain the symbol "o" in caption*
Caption amended by "adequate sufficiency (o)".